# Developing and Implementing a Culturally Consonant Treatment Fidelity Support Plan with the Apsáalooke Nation

**DOI:** 10.3390/ijerph20216989

**Published:** 2023-10-28

**Authors:** Shannen Keene, Sarah Allen, Alma Knows His Gun McCormick, Coleen Trottier, Brianna Bull Shows, John Hallett, Rae Deernose, Suzanne Held

**Affiliations:** 1Department of Human Development & Community Health, Montana State University, Bozeman, MT 59717, USA; luzning@hotmail.com (C.T.); brianna.bullshows@gmail.com (B.B.S.); suzanne@montana.edu (S.H.); 2Department of Family Life & Human Development, Southern Utah University, Cedar City, UT 84720, USA; sarahallen3@suu.edu; 3Messengers for Health, Crow Agency, MT 59022, USA; alma.mccormick@montana.edu; 4Petaluma Health Center, Petaluma, CA 94954, USA; johnhallett@phealthcenter.org

**Keywords:** treatment fidelity, relational accountability, community-based participatory research, chronic illness self-management, Indigenous, Indigenous research methods

## Abstract

Treatment fidelity remains underreported in health intervention research, particularly among Indigenous communities. One explanation for this gap is the lack of culturally consonant strategies listed in the National Institutes of Health (NIH) Behavior Change Consortium (BCC) treatment fidelity framework, the gold standard for understanding and measuring fidelity. This paper focuses on the development and implementation of a culturally consonant treatment fidelity support plan across two of the five BCC fidelity areas, provider training and treatment delivery, within a chronic illness self-management program for the Apsáalooke (Crow) Nation. Our team selected and adapted strategies from, and added strategies to, the BCC framework, that centered on relational accountability and the Apsáalooke culture. To be culturally consonant, we approached treatment fidelity as supporting Aakbaabaaniilea (Apsáalooke program facilitators) rather than monitoring them. This resulted in the development of a fifth treatment fidelity area: building and fostering relationships. We propose that fidelity to relational accountability is the foundation of successful programs in Indigenous communities. This suggests an important shift from tracking what was conducted in an intervention to prioritizing how things were conducted. We encourage others to view the BCC framework as a starting point in developing fidelity strategies that are consonant with local cultures.

## 1. Introduction

In the Apsáalooke (Crow) language, Bílaxpaake báaxiakuleetak means people are aware of each other and there is nothing hidden (see Table 1 for a list of Apsáalooke words and translations). Due to the close-knit relationships embedded in the Apsáalooke clan system, community members know each other’s character and integrity, and how they cope with positive and difficult life experiences. Over time, someone who follows through with what they say and who can then be trusted is viewed as Bílaxpaaktialeek. The Apsáalooke have always valued aligning words with action—or following protocol—in everyday life, including in ceremonies, spiritual practices, and caring for relatives and others. This has been referred to in Indigenous Research Methods (IRM) as relational accountability, or a methodology “based in a community context” that “demonstrate[s] respect, reciprocity, and responsibility” [1] (p. 7) with others to align words with actions. The practice of upholding relational accountability is of utmost importance when working with, and for, the Apsáalooke community.

In the 1990s, health behavior researchers began to speak of Bílaxpaaktialeek, those who align words with actions, as the Western concept of treatment fidelity. This concept reflects “the methodological strategies used to monitor and enhance the reliability and validity of behavioral interventions” [2] (p. 443). In 2004, the Treatment Fidelity Workgroup of the National Institutes of Health (NIH) Behavior Change Consortium (BCC) introduced a framework to increase the utilization of treatment fidelity strategies in health intervention research [2], building upon existing fidelity models [3]. The BCC framework addresses five areas: (1) study design (what researchers test), (2) provider training (what providers acquire), (3) treatment delivery (what providers teach), (4) treatment receipt (what participants understand during the intervention), and (5) treatment enactment (what participants use outside the intervention). Under each area, Bellg et al. [2] provide goals with corresponding strategies compiled from 15 NIH-funded behavior change studies [4]. After selecting a set of strategies (a protocol) for each BCC area, researchers can assess if the strategies were carried out, or how words and actions aligned. Treatment fidelity helps ensure changes in the dependent variable are due to the independent variable [2,5], improve statistical power and effect size [6,7], reduce study costs [2], and allow for a more accurate explanation of study outcomes [8].

Despite literature supporting the importance of establishing and maintaining fidelity standards, treatment fidelity remains underreported in health intervention research. In a review of health intervention research articles published between 1990 and 2000, Borrelli et al. [9] found 22% of researchers reported adherence to fidelity strategies in provider training, followed by treatment delivery (35%), treatment receipt (49%), treatment enactment (57%), and study design (80%). In a scoping review of behavioral adult tobacco treatment interventions published between 2006 and 2018, Salloum et al. [10] found researchers reported adherence to BCC strategies as follows: provider training (14%), treatment delivery (15%), treatment receipt (16%), treatment enactment (25%), and study design (68%). Other researchers have also found low levels of reporting among provider training and treatment delivery [11,12,13]. Low levels of reporting may be due to poor transparency of research protocols, unfamiliarity with the BCC framework, the lack of fidelity reporting requirements in the current CONSORT guidelines or journal manuscript guidelines, or the culture of non-reporting in standard disciplinary practices [9,10,12,13].

Treatment fidelity is also underreported in health intervention research with Indigenous communities. Part of the underreporting is likely due to the lack of culturally consonant fidelity strategies in the BCC framework since the strategies were derived from intervention research with non-Indigenous communities. Despite authors calling for the “flexible adaptation [of BCC strategies] according to setting, provider, and patient” [5] (p. S52), we have not seen literature regarding how researchers working in Indigenous communities have selected, adapted, and added strategies fidelity strategies based on their consonance with the local culture.

The purpose of this paper is to describe how our community-based participatory research (CBPR) partnership selected and adapted strategies from, and added strategies to, the BCC framework that centered on relational accountability and the Apsáalooke culture in two fidelity areas: (1) provider training and (2) treatment delivery. The strategies informed our treatment fidelity support plan which was implemented in the Báa nnilah (to share advice, usually through storytelling) Program—a health intervention designed to improve self-management of chronic illnesses (CIs) within the Apsáalooke Nation. Our treatment fidelity support plan was intended to support Aakbaabaaniilea (ones who give good advice), or Apsáalooke program facilitators, in leading the Báa nnilah Program.

## 2. Community Context 

The Apsáalooke or “children of the large-beaked bird” are “exactly in the right place” as stated by Chief Eelápuash (Sore Belly) [14] (p. 118). In a time of prayer and fasting, Chief Shíipdeetash (No Vitals) received spiritual guidance from the Creator in the form of a vision to search for and migrate to a place where a plant that “twinkled like the stars” grew. With their hearts and by faith, the nation traveled and after several generations found Ihchihchia (the sacred tobacco plant) at the foot of the Iisaxpúatahchee Isawaxaawúua (Bighorn Mountains), near the Baahpuuo Isawaxaawúua (Pryor Mountains) and Cheétiish (Wolf Mountains). Unfortunately, after finding this right place, the federal government forced them onto a reservation in present-day southeastern Montana. Repeated broken treaties between the federal government and the Apsáalooke Nation resulted in a reduction of the size of the reservation from 38 million acres to its current size of 2.25 million acres [15,16]. Today, there are six unique districts or communities across the reservation: Alúutaashe (Arrow Creek/Pryor District), Baleewaakeeáashe (Big Horn District), Iikooshtakáatbaatchaache (Mighty Few/Wyola District), Ashshipíte (Black Lodge District), Áshkualee (Center Lodge/Reno District), and Áashbacheeitche (Valley of the Chiefs/Lodge Grass District). In the Apsáalooke language, the name of each district describes the people living in those communities and their relationship to the land. 

Although more fortunate than many tribes, since the Apsáalooke are located on land that is sacred to them, the destructive impacts of colonization continue today in the form of many inequities, including health outcomes. This aligns with data showing that in Montana, American Indians/Alaska Natives with heart disease, diabetes, and cerebrovascular disease die 14, 11, and 10 years earlier than Whites, respectively [17]. Encompassing much of the Apsáalooke reservation, Big Horn County ranks second to lowest in health outcomes out of 47 Montana counties [18]. This county has almost triple (22,567 vs. 7500 per 100,000) the state’s rate of years of potential life lost before age 75 [18]. Real Bird et al. [19] have further explored the impact of historical and current loss on CI among the Apsáalooke. 

## 3. Intervention Context 

Since 1996, Apsáalooke community members and Montana State University faculty and students have partnered to develop, implement, and evaluate health programs based on community interests and strengths. In 2013, the partnership’s Community Advisory Board (CAB) identified support for community members with CIs as the next area of focus. To meet this need, our partnership developed, implemented, and evaluated the Báa nnilah Program. We evaluated the program using a randomized waitlist-controlled trial in which each of the two arms consisted of 10 groups of 11 Apsáalooke community members—one Aakbaabaaniilea and 10 participants—who were at least 25 years old and interested in improving their CI self-management. Aakbaabaaniilea were members of or had family in the district where they led their meetings (referred to as gatherings) and were seen as role models in their community for being engaged in their health. They led seven gatherings that covered topics including CI self-management, physical activity and healthy eating, historical and current trauma and resilience, patient–provider relationships, and healthy communication. From their personal experiences and community ties, they shared words of advice with and provided support to program participants. To start in a good way, Aakbaabaaniilea opened each gathering with a prayer and a healthy meal. This provided a strong foundation for the rest of the gathering which included an Apsáalooke story related to the topic of the gathering, a lesson, a sharing circle, and an activity completed in supportive partnerships. For more information about the Báa nnilah Program development, implementation, and evaluation, please see Iitáa Dáakuash et al. [20], Held et al. [21], and Hallett et al. [22].

Our team’s positionality contributed to how we engaged with the Apsáalooke community throughout the development and implementation of the Báa nnilah Program. Shannen Keene identifies as a first-generation biracial Filipino American woman who grew up in southern Indiana. Grounded in Filipino cultural values of kapwa (inner connectedness) and bayanihan (solidarity), she actively works to uplift the community’s voices and strengths. Sarah Allen is an Associate Professor and Department Chair of Family Life and Human Development at Southern Utah University. She is a white, female, colonial settler from Canada. Her research, teaching, and service have informed her approach to strength-based programs that build individual, family, and community health and well-being in culturally consonant ways. Alma Knows His Gun McCormick is a member of the Apsáalooke Nation and the Executive Director of Messengers for Health, an Apsáalooke non-profit organization. She is a passionate leader and a community activist who desires to see improved health and wellness among her people. Coleen Trottier is a member of the Blackfeet Nation and is also Turtle Mountain Chippewa. She completed her Bachelor of Science at Montana State University in Psychology and is currently on a path toward attending medical school. She works as a student researcher to help in the advancement of ways to improve health so that the approach to better health best serves the community in which it is carried out. Brianna Bull Shows is an Apsáalooke and Pikuni (Blackfeet) tribal member who grew up on the Apsáalooke reservation in Pryor, Montana. She identifies as a kaalishbaapite (grandmother’s grandchild). The knowledge that has been passed down through generations shaped her understanding of treatment fidelity. John Hallett is a community physician who is white and was raised in Wisconsin. He practices trauma-informed primary care with underserved communities in Northern California. He does this work because he believes that relationships and personal stories are central to health. Rae Deernose, Iitáa Dáakuash (Always Has a Good Place to Be), is a member of the Crow Nation and grew up in a five generation family in Benteen, Montana on the Crow Reservation. She received her bachelor’s and master’s degrees in Community Health from Montana State University. During graduate school, she joined Messengers for Health and focused her research on IRM. She also worked to further ignite an Apsáalooke methodology, Walk Story, in Crow Agency, Montana to incorporate methodologies such as making relatives, storytelling, humor, ceremony, language, and physical activity. Suzanne Held is a Professor at Montana State University. She has worked since 1996 as a non-Indigenous partner with the Messengers for Health program. Her interests are to work in partnerships to establish trust, share power, foster co-learning, and address community-identified health issues using strengths- and community-based approaches.

## 4. Methods

### 4.1. CBPR and IRM Approach

Our research approach is led by Apsáalooke cultural values and IRM. This approach resulted in conceptualizing fidelity as modeling Bílaxpaaktialeek because Aakbaabaaniilea are recognized and honored for their good character within the community. This meant doing what we said we would do within our partnership of Indigenous and non-Indigenous scholars as well as within our partnerships among the Aakbaabaaniilea, the Báa nnilah participants, and the Apsáalooke community. Leading with Apsáalooke values also aligns with IRM principles of relevance, reciprocity, respect, relationality, responsibility, and redistribution [1,23,24,25]. Collectively, these values and principles shaped how we showed up with a good heart, built trust between the research team and Aakbaabaaniilea, and developed and implemented the fidelity plan.

We referred to our treatment fidelity support plan as the Aakbaabaaniilea support plan and believed that supporting Aakbaabaaniilea would allow them to follow through with supporting program participants, which would allow the participants to follow through with actions to support their health. Aakbaabaaniilea were lay community health workers with varying levels of comfort and experience in leading groups. They also served many roles in their community and families, while still managing their CIs. Consequently, we wanted to be mindful of additional responsibilities required by the program and be intentional in how we provided one-on-one support to each Aakbaabaaniilea to lessen or alleviate the stress that the role placed on them. We did this to show respect for their time, knowledge, willingness to share with others, and trusted relationships within the community.

### 4.2. Aakbaabaaniilea Support Plan Development 

To develop the Aakbaabaaniilea support plan, team members discussed each of the five BCC areas and strategies and their consonance with the Apsáalooke culture over multiple team meetings. Through discussion and discernment, we determined which strategies to select, adapt, and add. Treatment fidelity strategies used in a tobacco cessation intervention for Alaska Native women also informed our work [26]. Below we describe the strategies that were implemented in our Aakbaabaaniilea support plan.

## 5. Results

### 5.1. Provider (Aakbaabaaniilea) Training

According to Bellg et al. [2], this fidelity area involves monitoring and enhancing training to ensure providers are prepared to deliver the intervention. To optimize training, the BCC suggested addressing four goals: (1) standardize training, (2) ensure provider skill acquisition, (3) minimize “drift” in provider skills, and (4) accommodate provider differences. We utilized various BCC strategies outlined by Bellg et al. [2] (shown in quotes in the text below) to meet each of these goals (see Table 2). We began by reconceptualizing the definition of this area to be providing support versus monitoring as the idea of monitoring the Aakbaabaaniilea was not a culturally appropriate approach. 

#### 5.1.1. Goal 1: Standardize Training

This goal intends to ensure that the research team provides training similarly across providers [2]. The majority of Aakbaabaaniilea (nine out of 10) “train[ed] together” in person “us[ing] standardized manuals” (the remaining Aakbaabaaniilea received one-on-one training due to scheduling conflicts). We “use[d the] same instructors for all providers”, which was the Executive Director of Messengers for Health. The training, which took the form of a two-day retreat, “us[ed] structured practice and role-playing” for Aakbaabaaniilea to practice facilitating each gathering. The CBPR team served in a supportive role to encourage them and answer any questions they had. For the retreat, to be respectful and consonant with the Apsáalooke culture, we brought together the Aakbaabaaniilea, their families, and the CBPR team as equals. We ate meals together, socialized before and after the training, and had a chance to deepen our relationships through laughter and conversation. We chose this format because it was congruent with the Apsáalooke cultural ways of reciprocity and hospitality through honoring and expressing appreciation for Aakbaabaaniilea and blessing them for their willingness to take part in the program. We wanted to let Aakbaabaaniilea know we cared about them by helping them feel comfortable, welcome, and special. This modeled how we wanted Aakbaabaaniilea to welcome participants to their gatherings.

We also selected and adapted various other BCC strategies to standardize training. We used a training manual—developed by our CBPR team—that included program gathering summaries and addressed topics such as how to support participants, be mindful of cultural communication patterns, maintain confidentiality, facilitate sessions, and prepare for gatherings. In other words, the “training [took] into account the different experience levels” of Aakbaabaaniilea. We also “design[ed] training to allow for diverse implementation styles” by encouraging them to show their personalities, use their own words, speak the Apsáalooke language (if applicable), and use humor. The final strategy that we utilized was to “ensure that providers meet a priori performance criteria”. To be culturally consonant, the performance criteria were based on lived experience, character, integrity, and respect in the community as positive role models. This has been referred to as reservation credibility—or rez cred. This is different from Western views of leadership, which emphasize education and professional experience, though some Aakbaabaaniilea had backgrounds in relevant fields including community health, social work, and nursing in addition to reservation credibility. 

#### 5.1.2. Goal 2: Ensure Provider Skill Acquisition

This BCC goal involves researchers “train[ing] providers to well-defined performance criteria” [2] (p. 447). The language of well-defined performance criteria did not resonate with the Apsáalooke members of our team. As a result, we focused on Aakbaabaaniilea feeling comfortable and confident in facilitating the intervention. To be consonant with the Apsáalooke culture, we did not “provide written exam pre- and post-training” as evidence of skill acquisition of the content. Instead, we had team members at the retreat who could provide real-time support and guidance. Also, to be consonant with the culture, the retreat ended with an honoring ceremony where an elder in the community (the Executive Director) provided each Aakbaabaaniilea with a certification of training completion. She publicly shared words of appreciation, encouragement, and support and described their dedication to the community and preparedness to be effective leaders of the Báa nnilah Program (“certify interventionists initially”). We also “conduct[ed] provider-identified problem solving and debriefing” by holding bi-monthly support calls and monthly meetings to review the seven gatherings and share advice. These calls and meetings were focused on Aakbaabaaniilea-identified topics. 

#### 5.1.3. Goal 3: Minimize “Drift” in Provider Skills

By optimizing this goal, researchers support providers in maintaining their skills across time [2]. In the Báa nnilah Program, we “provide[d] multiple training sessions” and “conduct[ed] regular booster sessions” with Aakbaabaaniilea. We did not “audio record and review” the training due to its inconsonance with the Apsáalooke culture. During the retreats, team members served in a supportive role while Aakbaabaaniilea practiced their role as a facilitator. This is a culturally consonant and relational version of the BCC strategy of “conduct[ing] in vivo observation”. Suggestions for strengthening facilitation were provided to the group versus individually, which is an appropriate manner for providing feedback. After the initial two-day retreat, we “conduct[ed] periodic meetings with providers”. Monthly meetings also provided a space for local professionals to lead trainings on topics that Aakbaabaaniilea were interested in such as healthy eating and mental health. Before the waitlist control group started, we held another two-day booster retreat focused on the three (of seven) gatherings that Aakbaabaaniilea identified as feeling the least comfortable and confident with facilitating based on a “self-report questionnaire”. Lastly, we ensured that Aakbaabaaniilea had “easy access to project staff for questions about the intervention”. In consonance with the culture, we fostered relationality by creating a welcoming, comfortable, and open discussion space for Aakbaabaaniilea to share stories of their experiences facilitating the program, ask questions and provide input in monthly meetings, training, bi-weekly support calls, and via text or phone. 

#### 5.1.4. Goal 4: Accommodate Provider Differences

This goal ensures that researchers adequately train providers with varying levels of skills and backgrounds [2]. While “giv[ing] all providers intensive training”, the Executive Director supported each Aakbaabaaniilea when role-playing each gathering (“have professional leaders supervise lay group leaders/paraprofessionals”). She is seen as a leader in the Apsáalooke community and someone with good character and integrity (Bílaxpaaktialeek). To further support provider differences, prior to intervention implementation, we asked Aakbaabaaniilea about the type of support they wanted from the CBPR team (e.g., weekly check-ins, phone calls, texts) via a self-report questionnaire. We then tailored our support to their needs (“use provider-centered training according to needs”) which included “us[ing] regular debriefing meetings”. 

We viewed the unique backgrounds and experiences of our Aakbaabaaniilea as a strength of our program. As such, we encouraged Aakbaabaaniilea throughout the training to use their own words, experiences, and stories to convey the meaning, intent, and content when delivering the gathering rather than reading from a script. This approach is congruent with the Apsáalooke cultural value of speaking good words from the heart that connects with the needs of the participants. This helped personalize the program delivery approach to reflect their personalities and best meet their participants’ needs.

Two additional ways provider differences were accommodated in culturally consonant ways were using the Apsáalooke language and the way feedback was provided to the Aakbaabaaniilea. The Executive Director, a fluent Apsáalooke speaker, worked one-on-one at trainings with an Aakbaabaaniilea who felt more comfortable leading the gatherings in the Apsáalooke language. This accommodation allowed him to understand his role more clearly. She also provided feedback to Aakbaabaaniilea by honoring their unique strengths at monthly meetings, while also addressing areas for growth with the entire group. As mentioned above, it is culturally appropriate to provide constructive feedback to a group versus to an individual to avoid shaming (“have inexperienced providers add to training by attending workshops or training programs”). We highlighted strengths such as expressing genuine concern for participants, communicating in the Apsáalooke language, and using humor to break the ice. Our CBPR team did not approach this fidelity area with the intention to “supervise” Aakbaabaaniilea as stated in BCC strategies. Instead, we supported their growth, encouraged them, and uplifted their inherent strengths.

### 5.2. Treatment Delivery 

By implementing strategies related to treatment delivery, researchers can ensure that providers deliver the intervention as intended based on the protocol [2]. Goals include: (1) control for provider differences, (2) reduce differences within treatment, (3) ensure adherence to treatment protocol, and (4) minimize contamination between conditions. Our partnership used diverse strategies to address each BCC goal (see Table 3).

#### 5.2.1. Goal 1: Control for Provider Differences

This goal involves selecting providers based on specific characteristics and assessing participants’ perceptions of the providers’ non-specific treatment effects (e.g., warmth, credibility, and empathy) [2]. As discussed above, to be consonant with Apsáalooke views on leadership, we “select[ed] providers for specific characteristics” based on their life experience with CIs, character, integrity, and respect in the community as positive role models. Another strategy under this goal is to “assess participants’ perceptions of provider warmth and credibility via self-report questionnaire”. To assess the warmth and credibility of Aakbaabaaniilea, we asked participants after they completed the intervention to rate the extent to which they felt that their Aakbaabaaniilea cared about or was concerned about them on a 4 point Likert scale (1 = not at all; 5 = very much). We also “monitor[ed] participant complaints” and “conducted a qualitative interview at [the] end of [the] study” with participants. 

#### 5.2.2. Goal 2: Reduce Differences within Treatment

This goal serves to “ensure that providers in the same condition are delivering the same intervention” [2] (p. 448). In our program, we “use[d] a scripted intervention protocol” and “provide[d] a treatment manual”. As mentioned above, we encouraged Aakbaabaaniilea to personalize their delivery using their own stories and experiences instead of reading verbatim from the manual. Although this approach may have increased variance among Aakbaabaaniilea, our CAB deemed this to be a culturally consonant approach, likely boosting the perceived warmth and credibility of Aakbaabaaniilea. 

#### 5.2.3. Goal 3: Ensure Adherence to Treatment Protocol

Often thought of as the heart of fidelity [27], this goal ensures that providers deliver the intervention according to the intended dose and intervention components. We evaluated the gathering implementation by having Aakbaabaaniilea “complete a behavioral checklist of intervention components delivered” at the end of each gathering. Aakbaabaaniilea were asked to put a check mark if they completed each of the five sections of the gatherings: (1) Welcome, Prayer, and Meal, (2) Story, (3) Introduction/Purpose of Gathering, (4) Supportive Partnerships Activity, and (5) Closing. This allowed us to “check for errors of omission and commission in intervention delivery”. We believed our strong CBPR partnership helped Aakbaabaaniilea feel comfortable in “reporting deviations from treatment manual content” on the checklist. The checklist also included questions about how comfortable and confident they felt leading the gathering on a 10 point Likert scale (1 = not at all; 10 = very) as another way to communicate if they needed additional support in delivering the program. 

The third author, the Messengers for Health Executive Director, conducted an in-person evaluation of a 10% random sample of the gatherings as an additional way to evaluate adherence. The gatherings were not video recorded as our CAB and community member staff felt that doing so would take away from the safe and confidential space that was created. The evaluation included identifying the extent to which each gathering component was implemented as intended (i.e., not completed, partially completed, or fully completed) and other factors that may have influenced the gathering (e.g., participant or environmental factors and Aakbaabaaniilea interactional style). She approached the evaluation as an active participant in the gathering, further honoring the relationship she had with Aakbaabaaniilea and the participants. If Aakbaabaaniilea felt unsure during the gathering, they felt comfortable openly asking for her advice and input. Aakbaabaaniilea modeled respecting that advice as their participants did for them. These activities were our culturally consonant adaptation of the BCC’s strategy to “randomly monitor [intervention sessions] for both protocol adherence and non-specific treatment effects”. 

We also added two new strategies to support treatment protocol adherence. First, the first author called and/or texted Aakbaabaaniilea before each gathering to check in and remind them of materials to bring to the gathering. This is another example of relational accountability. Second, we developed “how to prepare for each gathering” information sheets to meet the request from Aakbaabaaniilea for more organizational tools. These handouts included actions to take before, during, and after the gathering. 

#### 5.2.4. Goal 4: Minimize Contamination between Conditions

According to Bellg et al. [2], this goal ensures that the control group did not receive exposure to the intervention. During program development, we acknowledged that the Western concept of contamination did not align with the Apsáalooke cultural strength of sharing information with others. Rather than trying to prevent sharing between intervention and waitlist control conditions, we asked participants in the waitlist control group (n = 93) in a follow-up survey “Have you taken action for your health because of something you have heard or learned from a friend or family member who has attended the gatherings?” [28] (p. 8). This was conducted after the intervention group completed the intervention and prior to the waitlist control group taking part. If they indicated yes, we followed up with: “Please share what actions you have taken or changes you have made in your life because of what you heard or learned from a friend or family member who has attended the gatherings”. This was our adaptation of the BCC’s strategy, “conduct[ing] patient exit interviews to ensure that control subjects did not receive treatment”.

Our results reaffirmed the cultural value of relationality in that we found a high rate of sharing program content between the intervention and waitlist control groups as well as with the community at large. Eighty percent of waitlist control participants responded “yes” to the question of taking action to improve their health based on something they learned [28]. Although within a Western lens, this reduces the ability of researchers to assess program impacts, our partnership saw this as direct evidence the program was working as intended.

### 5.3. Building and Fostering Relationships

We added a new area to the BCC framework that reflects our CBPR team’s goals of building and fostering relationships that are grounded in trust, integrity, and genuine care and concern. With these intentions, researchers can work towards the following goals: (1) identify and act on support needs, (2) promote relational accountability across the research team, (3) support provider communication with participants, and (4) promote a culture of care. 

#### 5.3.1. Goal 1: Identify and Act on Support Needs

By working towards this goal, researchers can ensure that they are meeting the evolving needs of providers. In the Báa nnilah Program, this occurred through routine in-person and telephone check-ins with Aakbaabaaniilea and by careful observation of Aakbaabaaniilea. Upon learning about or noticing support needs, we followed through. For example, when we learned that an Aakbaabaaniilea needed support with transportation, program staff gave them a ride to the grocery store to pick up food for the opening meal and to their gathering. Also, we noticed that one of the Aakbaabaaniilea was holding the program manual close to their face, making it more difficult to facilitate the gatherings, and inferred that they were having difficulty reading the small print. We followed up one-on-one and after asking if a larger font size would help, printed and provided a new version of the manual. By creating a safe place and facilitating open dialogue for providers to share their unique needs, research teams can address needs and foster trust within their partnership. 

#### 5.3.2. Goal 2: Promote Relational Accountability across the Research Team

The purpose of this goal is to foster the research team’s active responsibility to support providers, understanding that every team member contributes to the program’s success. The Messengers for Health Executive Director attended many gatherings to show genuine care, answer questions, and help in whatever way Aakbaabaaniilea needed. For example, she brought more food to a gathering when an Aakbaabaaniilea realized they did not have enough food. This was especially important because they had invited the family members of a participant who had passed on. Montana State University staff, students, and program volunteers also attended gatherings. Some Aakbaabaaniilea also attended others’ gatherings to provide peer support, serving as a backup and complementing the efforts of the Aakbaabaaniilea leading. As a result, at least one team member supported each Aakbaabaaniilea in person at most of their seven gatherings. Indeed, post-program interviews with Aakbaabaaniilea revealed perceptions of timely and accessible support from the CBPR team, resulting in feeling secure, at ease, and not alone.

#### 5.3.3. Goal 3: Support Provider Communication with Participants

This goal serves to ensure that the research team supports providers in communicating key program information to participants in a culturally consonant and respectful way. After learning that some participants were not attending gatherings and that this impacted the comfort and confidence of Aakbaabaaniilea leading gatherings, research team members connected with Aakbaabaaniilea to support them by providing texts or making phone calls to their participants to remind them of gathering dates, times, and locations. One Aakbaabaaniilea preferred to text their participants. If we were unable to reach participants, we identified additional contacts who could share the information with them. We also communicated with participants when we modified gathering dates to reduce conflict with community events (e.g., basketball games or cultural gatherings), inclement weather, and deaths in the community. This reflects ongoing team support for flexible timelines as well as respect for the community and the Apsáalooke values. 

#### 5.3.4. Goal 4: Promote a Culture of Care

This goal intends to prioritize the providers’ well-being throughout intervention planning, implementation, and evaluation. Our CAB and Aakbaabaaniilea emphasized the importance of well-being based on their belief that we must first take care of ourselves to fully show up for others. Throughout the program, we sent supportive text messages to Aakbaabaaniilea expressing encouragement and gratitude. For example, we texted Aakbaabaaniilea: “As you end your week, remember to be gentle with yourself. You are doing the best you can. Please let me know if I can help you in any way”. Additionally, we integrated mindfulness activities into monthly meetings, encouraged self-care plans, and provided healthy recipe booklets based on Aakbaabaaniilea feedback. This care also extended to their family members when the team expressed care and concern for those who were sick or injured. Similarly, Aakbaabaaniilea reciprocated this care for the CBPR team by offering prayers for their well-being and safe travel through the often dangerous nighttime driving in rural Montana. This support within our partnership was congruent with the Apsáalooke cultural values of Diichikaatah (taking good care of yourself) and Báachiikitáalah (taking care of one another). Similarly, one Aakbaabaaniilea spoke of self-care as an integral part of “practic[ing] what you preach” because “being a mentor is being a role model and being able to give advice to others”, as shared by another Aakbaabaaniilea (Biadaahissaash/Wealthy Woman). They believed that by taking care of themselves and each other, they could “fully get the program out the way we want it to”. This approach was fundamental to fostering trusting relationships and implementing the support plan in a good way.

## 6. Discussion

There is consonance between the Apsáalooke concept of Bílaxpaaktialeek, those that align words with actions, and the Western research concept of treatment fidelity. Grounding our treatment fidelity approach in the Apsáalooke culture and relational accountability [1] led us to select and adapt various BCC strategies and to add a new area (Building and Fostering Relationships) to the five BCC areas. This made our approach fundamentally different from Western conceptualizations of treatment fidelity. We focused on strategies that centered on relationality and supporting Aakbaabaaniilea versus monitoring them. We walked side by side with Aakbaabaaniilea, actively seeking out and responding to the ways we could best support them. We believed that providing this ongoing support strengthened the capacity of Aakbaabaaniilea to lead the program and further strengthen the community’s capacity to improve their health and well-being. 

Our treatment fidelity approach is consistent with the Apsáalooke culture and IRM, as we prioritized building respectful and reciprocal relationships. We believe this was foundational to the program’s success in health promotion [1,23]. We approached providing support as a dynamic and individualized process where we asked for and responded to the needs of Aakbaabaaniilea. As Atkinson [29] states, we have a “responsibility to act with fidelity in relationship to what has been heard, observed, or learnt” (p. 10). This is a way to “show honor, consider the well-being of others, and treat others with kindness and courtesy” [30] (p. 86). We also encouraged Aakbaabaaniilea to uphold relational accountability to themselves by uplifting and honoring their needs, stories, and voices throughout the program [1]. As shared by Hawe, Shiell, and Riley [31], adopting a dynamic approach allows for flexibility of the form (i.e., the approach of each Aakbaabaaniilea) while the function of the intervention is standardized. Collectively, these strategies are congruent with conceptualizations of IRM as a way of thinking and knowing built upon context, content, and community [1,29].

The BCC strategies selected for our support plan were consistent with those used in health interventions among Indigenous communities. For example, many studies used standardized training materials and role-playing in training sessions [26,32,33,34], as well as treatment manuals during the intervention [34,35,36]. Kaholokula et al. [37] also balanced adhering to the protocol with the flexibility for providers to incorporate their own experiences and stories. To track program adherence, Aakbaabaaniilea completed behavioral checklists, a commonly used strategy across studies [26,36,38]. Additionally, we evaluated a 10% random sample of intervention gatherings using a standardized checklist as used by other researchers [34,35,38]. Similar to Canuto et al. [35], our evaluation checklist allowed the Messengers for Health Executive Director to rate implementation on a multi-point scale and capture contextual factors that may have impacted the gathering (e.g., health conditions, weather, deaths). These treatment fidelity strategies provided additional means to support Aakbaabaaniilea and foster accountability throughout the program. Although we selected similar BCC strategies as other researchers, our team took time to apply the strategies in a way that was consistent with the Apsáalooke culture, something we did not see described in existing literature. 

Centering Indigenous values and following Indigenous protocols does not necessarily mean the rejection of all Western methods and theories. Instead, it means that the community leads the determination of what should be selected, adapted, and added to be appropriate and beneficial [39,40]. Other scholars have discussed the important tension of bridging the demands of Indigenous community values and needs and Western research methods in ways that privilege tribal sovereignty and researcher-community relationships over academic concerns [39]. Scholars have shared how they created a “syncretic and blending of Indigenous and Western theories and practices” [41] (p. 18) in their CBPR partnerships. Our treatment fidelity support plan incorporates elements of both Indigenous and Western research methods as deemed appropriate by our community partners. For example, although we utilized the BCC strategy of a self-report checklist and evaluation, we approached the evaluation component in a supportive, culturally consonant, and compassionate way. We recognized how the foundations of Western research methods have been used as forces of colonization [28,42,43] and followed the leadership of our community partners regarding the most appropriate way to move forward through these tensions.

### 6.1. Limitations 

We experienced challenges during the implementation of the support plan. First, some Aakbaabaaniilea forgot to complete sections of the behavioral checklist used to track adherence to the gathering protocol, and two lost their checklist packet. This resulted in missing data, a common issue in fidelity research [44]. To minimize this risk, researchers can review or make copies of behavioral checklists regularly. Second, we did not provide definitions of “confident” and “comfortable” on the behavioral checklist. Without a uniform definition, responses could have varied among Aakbaabaaniilea. 

### 6.2. Implications for Practice and Future Directions 

Our development and implementation process can inform how future practitioners and researchers approach treatment fidelity in health promotion interventions within Indigenous communities. First, we recommend that communities lead the conversation in selecting, adapting, or adding fidelity strategies that are relational to their culture. Next, we recommend that practitioners and researchers actively engage providers/facilitators to personalize how they would like to receive support. Implementing these strategies in a way that upholds relational accountability and fosters respect, reciprocity, and responsibility is vital. Given the power of sharing in many Indigenous communities, it is also important to assess whether the treatment fidelity goal of “minimizing contamination” is appropriate within specific community contexts. We discussed the Western bias embedded within the construct of contamination elsewhere [28] and believe that community sharing is an indicator of a program working as intended, instead of as a limitation. Finally, we recommend that others provide approaches to Indigenizing the three BCC fidelity areas not addressed in this paper. Through meaningful partnerships with communities, researchers and practitioners can develop and implement culturally consonant treatment fidelity strategies to advance health promotion and equity. 

## 7. Conclusions

In this paper, we described how our CBPR partnership developed and implemented a culturally consonant Aakbaabaaniilea support plan for the Báa nnilah Program, a chronic illness self-management program for the Apsáalooke Nation. Throughout the process, we were grounded in relational accountability—our CBPR team and Aakbaabaaniilea actively followed through with what we asked of each other. We worked to uphold our relational accountability by reconceptualizing treatment fidelity as providing ongoing, tailored support that centered on Apsáalooke values. In this way, we used the BCC framework as a tool we could select from, adapt, and add to in order to better meet the Aakbaabaaniilea and community’s needs. We hope our approach supports other communities in tailoring treatment fidelity to their local cultures.

## Figures and Tables

**Table 1 ijerph-20-06989-t001:** Apsáalooke Words and Translations.

Apsáalooke Word *	Translation
Apsáalooke	Crow; children of the large-beaked bird
Bílaxpaake báaxiakuleetak	People are aware of each other’s character and integrity; there is nothing hidden
Bílaxpaaktialeek	Someone who follows through with what they say and who can then be trusted; those who align words with actions
Báa nnilah	To share advice, usually through storytelling
Aakbaabaaniilea	Ones who give good advice
Eelápuash	Sore Belly
Shíipdeetash	No Vitals
Ihchihchia	Sacred tobacco plant
Iisaxpúatahchee Isawaxaawúua	Bighorn Mountains
Baahpuuo Isawaxaawúua	Pryor Mountains
Cheétiish	Wolf Mountains
Alúutaashe	Arrow Creek/Pryor District
Baleewaakeeáashe	Big Horn District
Iikooshtakáatbaatchaache	Mighty Few/Wyola District
Ashshipíte	Black Lodge District
Áshkualee	Reno District
Áashbacheeitche	Valley of the Chiefs/Lodge Grass District
Kaalishbaapite	Grandmother’s grandchild
Iitáa Dáakuash	Always Has a Good Place to Be
Diichikaatah	Taking good care of yourself
Báachiikitáalah	Taking care of one another
Biadaahissaash	Wealthy Woman

* The Apsáalooke words are listed in order of appearance.

**Table 2 ijerph-20-06989-t002:** Behavior Change Consortium (BCC) Treatment Fidelity Strategies Used to Strengthen Provider Training in the Báa nnilah Program *.

BCC Goal	BCC Strategies Selected or Adapted	BCC Strategies Not Used
Standardize training.	Ensure that providers meet a priori performance criteria; have providers train together; use standardized training manuals/materials/provider resources/field guides; have training take into account the different experience levels of providers; use structured practice and role-playing; observe intervention implementation with pilot participants; use same instructors for all providers; design training to allow for diverse implementation styles.	Use standardized patients; videotape training in case there needs to be future training for other providers.
2. Ensure provider skill acquisition.	Observe intervention implementation with standardized patients and/or pilot participants (role-playing); conduct provider-identified problem solving and debriefing; certify interventionists initially (before the intervention) and periodically (during intervention implementation).	Score provider adherence according to an a priori checklist; provide written exam pre- and post-training.
3. Minimize “drift” in provider skills.	Conduct regular booster sessions; conduct in vivo observation or recorded (audio- or videotaped) encounters and review (score providers on their adherence using a priori checklist); provide multiple training sessions; conduct weekly supervision or periodic meetings with providers; allow providers easy access to project staff for questions about the intervention; have providers complete self-report questionnaire.	Conduct patient exit interviews to assess whether certain treatment components were delivered.
4. Accommodate provider differences.	Have professional leaders supervise lay group leaders/paraprofessionals; give all providers intensive training; use regular debriefing meetings; use provider-centered training according to needs, background, or clinical experience; have inexperienced providers add to training by attending workshops or training programs.	Monitor differential drop-out rates; evaluate differential effectiveness by professional experience.

* This table includes treatment fidelity strategies outlined by Bellg et al. [2] (p. 447).

**Table 3 ijerph-20-06989-t003:** BCC Treatment Fidelity Strategies Used to Strengthen Treatment Delivery in the Báa nnilah Program *.

BCC Goal	BCC Strategies Selected or Adapted	BCC Strategies Not Used
Control for provider differences.	Assess participants’ perceptions of provider warmth and credibility via self-report questionnaire and provide feedback to interventionist and include in analyses; select providers for specific characteristics; monitor participant complaints; conduct a qualitative interview at end of study.	Have providers work with all treatment groups; audiotape sessions and have different supervisors evaluate them and rate therapist factors.
2. Reduce differences within treatment.	Use scripted intervention protocol; provide a treatment manual.	Have supervisors rate audio- and videotapes.
3. Ensure adherence to treatment protocol.	Randomly monitor audiotapes for both protocol adherence and nonspecific treatment effects; check for errors of omission and commission in intervention delivery; after each encounter, have provider complete a behavioral checklist of intervention components delivered; ensure provider comfort in reporting deviations from treatment manual content.	Provide computerized prompts to providers during sessions about intervention content; audio- or videotape encounter and review with provider; review tapes without knowing treatment condition and guess condition.
4. Minimize contamination between conditions.	Conduct patient exit interviews to ensure that control subjects did not receive treatment.	Randomize sites rather than individuals; use treatment-specific handouts, presentation materials, manuals; train providers to criterion with role-playing; give specific training to providers regarding the rationale for keeping conditions separate; supervise providers frequently; audiotape or observe sessions with review and feedback.

* This table includes treatment fidelity strategies outlined by Bellg et al. [2] (p. 448).

## Data Availability

Data sharing is not applicable to this article as no new data were created or analyzed.

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
