# Peer review of "Developing and Implementing a Culturally Consonant Treatment Fidelity Support Plan with the Apsáalooke Nation"

_ijerph, 2023, doi:10.3390/ijerph20216989_

Round 1

Reviewer 1 Report

Comments and Suggestions for Authors

see my review, please

Reviewer 2 Report

Comments and Suggestions for Authors

The manuscript lacks clear elaboration on the research gaps within the existing literature, sufficient data collected and robust data analysis to test hypotheses proposed. I suggest the authors added some empirical data to strengthen the writing of the manuscript.  

Comments on the Quality of English Language

Just fine.

Reviewer 3 Report

Comments and Suggestions for Authors

This was a very well written and interesting article that explored in great depth the application of fidelity concepts to community based programs conducted with Indigenous communities. There were a number of elements I really liked about this paper:

·       The findings extended the notion of treatment fidelity in considering fidelity to relationship development which was a nice theoretical and model development contribution.

·       There was terrific detail on the relationship development and implementation stages described in the article. This level of detail is often rare in academic publications and should be of great value to policy makers and practitioners interested in this topic.

·       The paper highlighted some of the tensions of different theoretical orientations in research and community based practice.

I don’t have any suggested changes I think the paper is ready for publication.

My only suggestion to the authors is that in future work they look at some of the work on implementation fidelity from a complex system perspective. This is a seminal paper in that literature related to health promotion:

https://pubmed.ncbi.nlm.nih.gov/19390961/

Penny Hawe and colleagues have written extensively on this topic in health promotion. We have used this work ourselves to consider fidelity concepts in health promotion practice at a settings and policy based level. Some of the theorizing on implementation fidelity and the need to consider process rather than content fidelity has been covered in systems thinking approaches to health promotion.

I don’t think this work is necessarily going to improve this paper as your conclusions are somewhat similar anyway. But if you are not familiar with this literature it may help further your thinking around fidelity and adaptation.

Reviewer 4 Report

Comments and Suggestions for Authors

Very Respected Authors,

After carefully reading your manuscript I have some suggestions. The objective  is defined too broadly. The manuscript is not clear. I suggest to choose one medical problem: obesity, mental health, AIDS, Covid-19 and than implement the specific approach. The method has to be clear, too. What type of study you done? 

Round 2

Reviewer 2 Report

Comments and Suggestions for Authors

The authors addressed well the concerns.

Reviewer 4 Report

Comments and Suggestions for Authors

I agree to accept in this form.